# The Role of Matrix Proteins in Cardiac Pathology

**DOI:** 10.3390/ijms23031338

**Published:** 2022-01-25

**Authors:** Katie Trinh, Sohel M. Julovi, Natasha M. Rogers

**Affiliations:** 1Centre for Transplant and Renal Research, Westmead Institute for Medical Research, Westmead, NSW 2145, Australia; klam8836@uni.sydney.edu.au (K.T.); sohel.julovi@sydney.edu.au (S.M.J.); 2Faculty of Medicine and Health Sydney, School of Medical Sciences, The University of Sydney, Sydney, NSW 2006, Australia; 3Renal and Transplantation Medicine, Westmead Hospital, Westmead, NSW 2145, Australia

**Keywords:** extracellular matrix proteins, cardiac, myocardial infarction, pressure overload, left ventricular hypertrophy, pulmonary hypertension

## Abstract

The extracellular matrix (ECM) and ECM-regulatory proteins mediate structural and cell-cell interactions that are crucial for embryonic cardiac development and postnatal homeostasis, as well as organ remodeling and repair in response to injury. These proteins possess a broad functionality that is regulated by multiple structural domains and dependent on their ability to interact with extracellular substrates and/or cell surface receptors. Several different cell types (cardiomyocytes, fibroblasts, endothelial and inflammatory cells) within the myocardium elaborate ECM proteins, and their role in cardiovascular (patho)physiology has been increasingly recognized. This has stimulated robust research dissecting the ECM protein function in human health and disease and replicating the genetic proof-of-principle. This review summarizes recent developments regarding the contribution of ECM to cardiovascular disease. The clear importance of this heterogeneous group of proteins in attenuating maladaptive repair responses provides an impetus for further investigation into these proteins as potential pharmacological targets in cardiac diseases and beyond.

## 1. Introduction

The extracellular matrix (ECM) has long been known as fundamental to the cellular configuration and regulation of function within organs, providing important ontogenetic signaling, postnatal structural integrity and remodeling changes in response to (patho)physiological stressors. The number of identified proteins within the ECM has increased substantially in the last two decades, in keeping with an expanding body of work that characterizes their role in health and disease, both preclinically and in human studies (Figure 1). Remodeling of the ECM typically accompanies the development of fibrosis that is deleterious to organ function. However, an understanding of ECM composition and function, and the capacity to modulate expression of ECM proteins, may hold the key to new therapeutic and regenerative opportunities in the heart and indeed all solid organs. A range of spatial and temporal morphologic changes occur in the heart during postnatal development as the heart adapts to the changed physiological environments imposed by ex-utero existence and growth. Mutations in several proteoglycans, collagens, fibrillins and fibronectins demonstrate embryonic or perinatal mortality [1,2], suggesting a fundamental role for these proteins in cardiovascular morphogenesis.

This review focuses on new findings regarding the role of non-structural ECM proteins in postnatal cardiac function and dysfunction, with an emphasis on the relevance to human cardiovascular pathology, ranging from myocardial infarction to hemodynamic overload. The current portfolio of well-characterized matrix proteins continues to expand, and increasingly sophisticated animal models have led to an improved understanding of their cellular origins and functional diversity in the heart. Well-known ECM proteins including collagen [3,4] and fibrillin [5] have already been extensively described in the literature, and therefore, we have directed our commentary to proteins with roles that are less well-defined.

## 2. Thrombospondins

Thrombospondins (TSPs) are a family of matrix glycoproteins with structural and oligomerization similarities and differences that were well-characterized in other reviews [6,7,8]. Cardiac expression of TSPs is constitutive in the ECM, albeit at low levels. They are fundamental to organogenesis but also increase significantly with injury, signifying a crucial role for TSPs in the reparative response. TSP-1 [9], -2 [10], -3 [11] and -4 [12] are all found in the context of cardiac remodeling.

TSP1 is protective after myocardial infarction (MI) [13] and in animal models of pressure overload (transverse aortic constriction, TAC) [14]. Murine infarcts demonstrate upregulated TSP1 expression at the injury border, and TSP1-null mice show sustained pro-inflammatory cytokine/chemokine secretion accompanied by enhanced macrophage and myofibroblast infiltration, which facilitate remodeling [13]. TSP1-null mice, compared to controls, also exhibit ventricular hypertrophy and late dilatation following pressure overload, with cardiomyocyte degeneration characterized by sarcolemmal disruption [14]. Other featured dysregulated responses include increased myofibroblast infiltration, matrix metalloproteinase (MMP) 3/9 activity and reduced transforming growth factor (TGF)-β activity. TSP1 is a known inhibitor of angiogenesis but also preserves the cardiac matrix through inhibition of MMP activity and by facilitating activation of latent TGF-β [8]. TSP1 also contributes to the right ventricular (RV) pathology as TSP1-null mice are protected from hypoxic stress, thus developing less RV hypertrophy and arteriolar thickening compared to control animals [15]. Further work identified marked upregulation of TSP1 expression (mRNA and protein) in the lung parenchyma and vasculature of patients with pulmonary hypertension [16], and increased plasma TSP1 levels have been found to correlate positively with the mean pulmonary artery pressure [17].

The majority of research into cardiac pathology has focused on the role of TSP1; however, TSP2 is required to maintain the integrity of the ECM. Microarray analysis of hypertrophic, failing hearts from Ren-2 (spontaneously hypertensive) transgenic rats demonstrated upregulated TSP2 expression. TSP2KO mice were unable to tolerate angiotensin II (ANGII) infusion (a standard model of cardiac hypertrophy that is independent of changes in systolic blood pressure), with significant mortality due to hemorrhagic cardiac rupture and subsequent pericardial tamponade [10]. This feature coincided with ultrastructural changes in mitochondria and concurrent overexpression of MMP 2 and 9. TSP2KO mice also demonstrate age-related cardiomyopathy, which could be prevented by adeno-associated virus gene transfer of TSP2 [18]. TSP2-null mice are more susceptible to cardiac rupture 48 h after MI [19]. The presence of TSP2 also protects against viral myocarditis, possibly due to depressed regulatory T cell (Treg) infiltration into the affected myocardium [20]. An elevated TSP2 transcript has been demonstrated in patients with LV hypertrophy and a concurrent depressed ejection fraction [10], and high TSP2 levels correlate with poor prognoses, including mortality, in heart failure [21,22]. There are significantly fewer publications on the remaining thrombospondins: TSP3 overexpression limits integrin expression and promotes sarcolemmal instability, leading to an exacerbation of cardiac injury [11]. TSP3KO mice are protected from TAC-mediated pressure overload. TSP4 is also rapidly upregulated in response to ANGII infusion or MI [23], and TSP4KO mice develop aggravated LV hypertrophy and fibrosis, as well as aortic aneurysms in response to pressure overload [24].

Three TSP single-nucleotide polymorphisms (SNPs) are associated with an increased risk of MI. Of these, an SNP in the coding region of TSP1 (N700S) confers an >8-fold increased risk of MI in Caucasians [25], but it is found at a relatively low frequency. This amino acid change is thought to reduce calcium-binding properties and increase platelet aggregability. The TSP4 SNP (A378P) is present at a high frequency in Caucasian populations and is associated with a lower rate of MI [26,27]. An SNP in the 3′ untranslated region of TSP2 is protective against MI, possibly through differences in RNA-binding proteins and overall TSP2 expression [28].

## 3. Matrix Metalloproteinases (MMPs)

The heart responds to injury through the elaboration of matrix proteins and subsequent ECM remodeling, leading to changes in ventricular geometry. Multiple different cell types contribute to the synthesis of ECM: injured parenchymal cells that may undergo phenotypic transformation under the influence of the microenvironment, infiltrating or resident inflammatory cells that are activated by neurohormonal or chemokine/cytokine release, and tissue-based fibroblasts all provide substrates for scar formation, regardless of the organ. This process requires both matrix metalloproteinases (MMPs) and their tissue inhibitors (TIMPs). MMPs are zinc-activated proteases, secreted in inactive forms, and can be detected in soluble and membrane-bound forms [29]. Of the 23 that are now described in humans (with different portfolios in mice), only a subset are expressed in the myocardium, with varying affinities for proteins as substrates [30], including ECM (type-4/5 collagen, laminin, fibronectin, elastin), other MMPs, bioactive peptides and growth factors. The structure and function of MMPs have been extensively reviewed in previous publications [31,32,33,34].

MMP-1, -2, -3, -7, -8, -9, -12, -14 and -28 expression are upregulated following MI [34,35,36], with a broad range of cells responsible for their elaboration. Several (particularly MMP-1, -2, -9) are crucial to post-injury remodeling [37], and all have been explored as potential markers of cardiovascular disease and post-MI outcomes. MMP-1 studies are complicated by dual isoforms and lack of significant homology between mice and humans; however, MMP-1 is elaborated predominantly by leukocytes, fibroblasts and endothelial cells. Levels in humans post-MI have been shown to correlate negatively with end-systolic volume and positively with ejection fraction [38]. MMP-2 levels mirror those of MMP-1 post-MI, with production by both cardiomyocytes and fibroblasts [39], although MMP-2 is constitutively active and crucially regulates tissue turnover under homeostatic conditions. The trajectory of MMP-9 elevation is compressed, although it has been shown to correlate with LV dysfunction [40] and cardiovascular mortality independent of other inflammatory markers [41]. MMP-2 and -9 feature strongly in the cardiomyopathic literature as they demonstrate an affinity for denatured ECM proteins, including collagen, fibronectin and laminin [30]. MMP-2-null mice were protected from cardiac rupture following MI, as were control mice treated with the MMP-2 (and MMP-9) inhibitor TISAM [(2R)-2-[5-[4-[ethyl-methylamino]phenyl]thiophene-2-sulfonylamino]-3-methylbutyric acid], despite no differences in infarct size [42]. This protection was associated with decreased leukocyte infiltration (particularly neutrophils and CD68+ macrophages) and preservation of the ECM’s structural integrity. The beneficial effect of MMP inhibition has been recapitulated in larger animal models [43,44,45], with a reduced infarct size even when pharmacotherapy is administered several days post-injury.

MMP transcription (particularly MMP-2 and -9) is also increased following exposure to exogenous ANGII, with upregulation of JAK-STAT [46] and NF-κB [47] in response to pro-inflammatory cytokines interleukin (IL)-1β and tumor necrosis factor (TNF)-α. Endothelin-1 (ET-1), which is strongly implicated in the pathogenesis of LV dysfunction in pressure-overload models, also leads to increased MMP levels [48] due to the pleiotropic nature of downstream protein kinase C activation [49]. MMP inhibition is effective at attenuating LV remodeling following pressure overload [50,51]. MMP-2-null mice are protected from TAC-induced LV hypertrophy [52], as are mice with transgenic overexpression of MMP-1 [53]. MMPs are less well-studied in models of RV pressure overload. MMP-1 [54] and MMP-2 [55] expression are upregulated in the vasculature, although these findings do not definitively demonstrate a causative role for MMPs. These findings have been corroborated by microarray studies of monocrotaline-treated rodent lungs (monocrotaline induces pulmonary arterial endothelial cell damage, arteritis and subsequent pulmonary hypertension), which revealed differential expression of ECM-related genes [56].

## 4. Osteopontin

Osteopontin (OPN, also known sialoprotein 1) is a non-collagenous protein present in the bone matrix and synthesized predominantly by osteoblasts (where it regulates the response of bones to external stressors) and mesenchymal stem cells. It is found in organs with and without an extensive ECM, in inflammatory disease (including multiple sclerosis [57] and rheumatoid arthritis [58]) and malignancy [59,60], suggesting both signaling and structural roles. The biological activity of OPN is modulated further by MMPs, and as its structure contains three cleavage sites, additional proteolytic processing can alter the functionality ([61,62], comprehensively reviewed in [63]). Indeed, OPN and MMPs co-localize at sites of wound healing, supporting the theory of a specific in vivo role for OPN fragments.

In cardiovascular-related studies, cardiomyocytes re-express OPN in response to chronic pressure overload [64], with additional identification localized to the perivascular space [65] and interstitium [65,66]. Cardiac expression of OPN was increased in a transgenic rat model of ANGII-dependent cardiac hypertrophy [67] and in wild-type (WT, C57BL/6) mice infused with ANGII [66] or subjected to TAC [65]. In further work using a mouse model of TAC-induced pressure overload, treatment with an OPN RNA aptamer at the time of TAC surgery to block downstream signaling prevented maladaptive cardiac remodeling and preserved cardiac function [68]. Treatment two months *after* TAC surgery also reversed cardiac hypertrophy, fibrosis and cardiac dysfunction by downregulation of phosphatidyl inositol-3-kinase and Akt phosphorylation, and reduced expression of additional ECM proteins.

Findings in OPN-null mice have varied extensively, although the dichotomy does not appear to be dependent on the model of ventricular overload. A blunted hypertrophic response has been demonstrated in the setting of chronic pressure overload [69], associated with reduced phosphorylation of several signaling molecules known to be involved in the hypertrophic response, including p38, Akt and c-Jun. OPN-null mice infused with ANGII had a reduced hypertrophic and fibrotic response when compared to control mice [70]. Similarly, Matsui et al. [66] showed that OPN-null mice infused with ANGII had an almost absent fibrotic response, with no significant change in LV hypertrophy. However, this was associated with decreased systolic function and increased dilatation. Worsening cardiac parameters in OPN-null mice, particularly collagen deposition and myofibroblast infiltration, have been recapitulated more recently [68], suggesting that OPN is indeed crucial for compensatory cardiac remodeling. The exact mechanism remains undefined, but OPN-null mouse cardiac fibroblasts demonstrated impaired adhesion to a variety of ECM proteins, and this capacity was partially restored by the addition of recombinant OPN when compared to wild-type cells [70]. Contributory roles of iNOS, calcium handling-related protein and cardiac cell apoptosis in the mechanism of OPN-mediated effects have been excluded [66]. Recently, it has been demonstrated that thrombin-cleaved, not-full-length OPN induces collagen expression by cardiac fibroblasts [65]. Syndecan-4, a transmembrane heparin sulfate proteoglycan, protects OPN from thrombin-mediated cleavage in the early phase of cardiac remodeling in response to pressure overload. In the late phase of remodeling, syndecan-4 shedding from the cell surface renders it unavailable for OPN binding, allowing for the production of thrombin-cleaved OPN, which drives cardiac fibrosis [65].

OPN is upregulated in the infarcted myocardium, with an early peak in the days following MI [71,72]. Production has been localized to macrophages in the infarct marginal zone [72]. The IL-10-STAT3-galectin 3 axis is essential for the activation and polarization of OPN-producing macrophages in the infarcted myocardium [73]. Studies suggest a cardioprotective role for OPN following ischemic insults [69,70], and OPN-null mice developed greater cardiac dilatation following MI, which was associated with reduced collagen production [71]. OPN-null mice subjected to repeated ischemic-reperfusion injury developed small, non-transmural infarctions and ventricular dysfunction due to cardiomyocyte loss, compared to WT controls [74]. OPN-null mice had reduced expression of tenascin-C (TN-C), MMP-9, MMP-12 and TIMP-1, but higher expression of MMP-13.

OPN may also have clinical utility as a biomarker. Patients with aortic stenosis and higher OPN plasma levels have lower left ventricular mass regression following surgical aortic valve replacements, suggesting that it may be predictive of LVH reversibility [75]. Pharmacological blockade of the ANGII receptor and HMG CoA reductase has been shown to reduce plasma OPN levels [76], implying its role in vascular remodeling. This may help identify patients who are at risk of developing heart failure or who may benefit from a surgical aortic valve replacement. Plasma OPN is also correlated with the coronary artery disease burden [77,78], as well as the cardiometabolic risk in the context of diabetes mellitus [79,80].

## 5. Periostin

Periostin is a multimodular matrix protein composed of a signal peptide (for secretion), C-terminal region (to interact with other ECM proteins), cysteine-rich module (for multimer formation) and fasciclin-like domains (for integrin interaction) [81,82]. Interestingly, in human periostin, the latter domains contain vitamin K-dependent γ-carboxyglutamic acid residues, which are features shared by factors involved in the coagulation cascade (factors II, VII, IX, X, proteins C&S) and bone-associated osteocalcin. Periostin plays both structural and organizational roles in the ECM, particularly collagen assembly, in addition to signaling functions to modulate cell behavior [83,84]. Periostin is preferentially expressed in the periosteum (hence its name) and in tissues with mechanical stress (including heart valves [85]). It is upregulated during cardiac development, although expression is limited to embryonic fibroblasts and pericardial cells, not cardiomyocytes [86,87]. Periostin is largely absent in the adult heart until it is re-expressed by fibroblasts within the cardiac extracellular matrix following MI [88]. TAC-induced pressure overload in WT mice also stimulates robust re-expression of periostin [88,89] within the cardiac interstitial space.

Under homeostatic conditions, periostin-null mice do not exhibit any change in cardiac morphology or function compared to WT control animals. However, the absence of periostin following MI results in greater mortality at 10 days post-injury due to LV wall rupture [88]. Surviving periostin-null mice demonstrated reduced fibrosis, scar formation and inflammatory (CD68+) cell recruitment, which correlated with improved functional recovery compared to WT mice. Administration of periostin peptide into the pericardial space improved the ejection fraction and myocardial contractility [90]. Similarly, TAC-induced pressure overload reduced collagen accumulation and slowed the progression of cardiac hypertrophy. This correlated with maintained ventricular performance compared to WT mice. Alterations in cardiac remodeling are in keeping with periostin’s known architectural interactions with other ECM proteins and association with TGF-β activity [86]. The other hypothesized mechanism is that periostin modulates the influx and characteristics of reparative (mesenchymal) cells following injury, akin to neoplastic cells that secrete periostin to expedite the metastatic capability and phenotypic transition [91,92]. Gene expression profiles in periostin-deficient fibroblasts are markedly different, particularly in relation to ECM and cell adhesion properties [88]. In vitro, cardiomyocytes fail to effectively attach to periostin-deficient fibroblasts compared to WT cells. Interestingly, restoration of the periostin protein to periostin-deficient fibroblasts did not enhance cellular adhesion, suggesting inherently different properties of periostin-deficient fibroblasts.

However, while periostin is necessary for scar formation, overexpressing transgenic mice did not develop a greater hypertrophic or fibrotic response to TAC or MI, and functional ventricular decompensation remained unchanged. This may be due to the FVB genetic background of these mice, which is less susceptible to cardiac decompensation compared to the C57BL/6 strain. These results suggest that periostin is necessary but not sufficient in the remodeling and fibrotic response.

Developmentally, periostin-null mice demonstrate ultrastructural valvular and architectural abnormalities, reflected in increased postnatal mortality [86]. These findings appear relevant to human valvular disease, as explanted pediatric valves demonstrate reduced periostin expression in concert with disorganized ECM. Samples from adults with valvular heart disease (atherosclerotic and rheumatic) demonstrated increased periostin expression (by IHC staining), which was secreted by myofibroblasts and CD14+ macrophages, although prolapsed valves showed no change [93]. In keeping with patterns in valvular pathology and small animal models, periostin is upregulated in a failing myocardium [94] and following MI [95].

## 6. SPARC

Secreted protein that is acidic and rich in cysteine (SPARC, also known as osteonectin or BM-40) modulates the ECM turnover in part through its effects on collagen synthesis, extracellular proteases and growth factors. SPARC is well-characterized in models of cutaneous healing, and SPARC-null mice demonstrate a reduced dermal type-1 collagen content (although increased type-4) and high molecular-weight complexes [96]. This protein also plays an equally important role in post-MI repair and responses to LV pressure overload. SPARC is pivotal to collagen assembly within the extracellular matrix and has been reported to play a crucial role in preserving the ventricular integrity after MI.

Expression of SPARC is increased after MI and is both spatially and temporally related to the formation of a fibrous scar, including α-smooth muscle actin (SMA) expression and CD45+ leukocyte infiltration [72]. Absent SPARC was associated with increased risk of myocardial rupture, and greater myocardial disruption in surviving mice. Histologically, this was associated with disordered, poorly assembled and immature collagen fibril deposition. However, the literature describes divergent results regarding the functional consequence of SPARC deletion following MI, with groups demonstrating significant early differences [97] or none [98], which may be accounted for by different strain backgrounds. However, studies all show increased mortality due to LV failure and/or rupture (albeit with a similar infarct size) secondary to failed scar formation, the fibroblast phenotype and altered TGF-β activity. SPARC binds to TGF-β receptor type-2 in the presence of TGF-β, and inhibition of SPARC expression blunts ρ-SMAD2/total SMAD2 expression in fibroblasts. Improved scar formation can be rescued with exogenous administration of SPARC or TGF-β [98].

Differences in macrophage presence have also been shown, suggesting that SPARC regulates the reparative response via the innate immune system [98,99]. SPARC also modulates Treg activity, which are recruited and clonally expand within the myocardium following ischemia-reperfusion injury. Transcriptomic analysis of heart Tregs demonstrated differential expression of SPARC, which was incidentally upregulated by IL-33. The importance of SPARC for regulating injury/repair processes post-MI was confirmed by additional experiments depleting CD25+ Tregs and concurrent adenoviral-mediated global overexpression of SPARC. Overall survival was improved with the overexpression of SPARC, attributable to few cardiac ruptures [100]. Absent SPARC also mediated changes in vascular permeability, leading to increased permissiveness of transmigratory capacity. This, in turn, was fundamentally due to a lack of glycocalyx integrity. Increased leukocyte infiltration, particularly pro-inflammatory Ly6C^hi^ monocytes and neutrophils in SPARC-null mice, was demonstrated in viral myocarditis, leading to increased cardiomyocyte injury, higher mortality and accelerated development of cardiomyopathy [101]. Vascular leakiness was restored by exogenous SPARC administration. However, repetitive SPARC injections could not rescue cardiac mortality in null mice, despite mitigating inflammation, and there were persistent differences in QTc times in SPARC-null mice compared to control mice during infection [101].

Pericytes derived from the adventitia of large vessels have been investigated for the therapeutic potential of MI [102,103]. Conditioned media from human pericytes identified SPARC as the bioactive matricellular protein that regulated the proliferation and migration of cardiac stromal cells [104]. SPARC expression was upregulated in response to hypoxia and starvation in platelet-derived growth factor (PDGF)-α positive cells. Silencing of SPARC reduced pericyte collagen production. In the same study, SPARC correlated with creatine kinase (CK) levels post-MI, and immunohistochemical examination revealed expression in interstitial and vascular cells. Interestingly, a subset of patients, regardless of MI size, failed to upregulate their SPARC levels. The clinical implications of this remain unclear. SPARC levels are increased in patients with coronary artery disease [105,106] and predict adverse cardiovascular outcomes in moderate to severe heart failure [107] but have not proved useful in distinguishing patients with acute MI [108,109].

Myocardial SPARC expression is also upregulated in response to TAC-induced pressure overload, with cellular secretion localized to fibroblasts rather than cardiomyocytes. This pattern mimics other ECM proteins relevant to cardiac injury and repair. The myocardial collagen content was less pronounced in SPARC-null mice compared to control mice [110]. This was associated with less diastolic dysfunction in SPARC-null mice, despite a similar rise in LV mass. Additionally, SPARC-null mice produced a disproportionately low level of insoluble collagen relative to soluble collagen when compared to control animals, suggesting that SPARC is involved in the development of mature cross-linked collagen fibrils. SPARC may mediate this effect by controlling type-1 collagen processing and altering collagen interactions with cardiac fibroblast cell surfaces [111].

Myocardial macrophages have been identified as a source of SPARC in the pressure-overloaded myocardium [112,113]. An increase in macrophage-derived SPARC and total collagen production is detected from one week in TAC-treated hearts, followed by an increase in insoluble collagen production and associated increase in myocardial stiffness at two weeks [112]. A recent study highlighted the importance of bone marrow-derived SPARC in the fibrotic response [113]. SPARC-null mice transplanted with wild-type bone marrow and subjected to LV pressure overload produced a similar fibrotic response to that of wild-type mice, and had a greater infiltration of cardiac macrophages when compared to wild-type mice transplanted with SPARC-null bone marrow.

SPARC has also been implicated in modulating the fibrotic response of the pressure-overloaded RV. SPARC expression was increased in the hypertrophied RV of rats with monocrotaline-induced pulmonary hypertension [114]. Pulmonary artery banding in a feline model induced RV chronic pressure overload within two weeks [115], and this was associated with an increase in the total collagen content. However, the insoluble collagen content did not increase until four weeks, and this correlated with an increase in SPARC expression in RV fibroblasts.

## 7. Tenascin C

Tenascins comprise a family of four homologs that are unique to vertebrates. These proteins share common, sequential motifs (amino-terminal heptad repeats, epidermal growth factor (EGF)-like repeats, fibronectin type-3 domain repeats and a carboxyl-terminal fibrinogen-like globular domain).

Tenascin-C (TN-C) is sparsely expressed in the normal adult myocardium, but reappears in some inflammatory situations such as MI [116], myocarditis [117], rheumatic heart disease [118] and hypertensive heart disease [119] (and extensively reviewed in [120]). In a mouse model of permanent coronary artery ligation, TN-C expression is highest at the border zone between necrotic and intact myocardium [121]. Previous studies have suggested that fibroblasts at the border zone begin to secrete TN-C 24 h after MI, and the expression peaks at three days [122]. TN-C is overexpressed by cardiac fibroblasts within the injured myocardium, which has provided a potential target when using nanoparticles [123]. Co-localization studies have attributed protein production to infiltrating macrophages. Indeed, decreased infiltration of CD11b+F4/80+ macrophages has been demonstrated in TN-C-null mice compared to controls (but not other infiltrating inflammatory cell populations including neutrophils or Ly6C+ monocytes), with a phenotypic switch in favor of M1 polarization [121]. These findings have been replicated in further studies, with TN-C levels peaking five days following MI [124]. In keeping with these findings, TN-C overexpression did not result in cardiac abnormalities at the baseline, but mice demonstrated augmented inflammation and increased mortality following MI [125].

TN-C deficiency mitigated cardiac damage and dysfunction with decreased infarct size and myocardial fibrosis, as well as functional improvements to the ejection fraction and fractional shortening. The authors also demonstrated decreased ROS-mediated cell death and inflammation via the NLRP3 inflammasome, and suppression of this pathway was via TLR4 and NF-κB. Beyond this, TN-C-null mice are protected from autoimmune myocarditis through lower Th17 cellular infiltrates. Dendritic cells stimulated with TN-C produce IL-6, IL-1β and GM-CSF, which preferentially promote Th17 differentiation and require intact TLR4 signaling [126].

Absent TN-C expression was also protective in a TAC mouse model, with null mice demonstrating reduced fibrosis, attenuated hypertrophy and preserved cardiac function associated with reduced MMP-2 and MMP-9 expression [127]. Cardiac fibrosis and inflammation were similarly attenuated in TN-C-null mice subjected to ANGII infusion [128]. In vitro studies showed that TN-C enhanced the migration and pro-inflammatory/pro-fibrotic function of macrophages via integrin αVβ3/FAK-Src and NF-κB [128]. However, conflicting results were seen in a subsequent study in which TN-C-null mice with a pressure-overloaded myocardium (induced by ANGII infusion or abdominal aorta constriction) demonstrated an exaggerated inflammatory response and reduced cardiac function compared to controls [129].

Clinically, TN-C may be a useful biomarker of cardiovascular disease or event burden. Serum TN-C was increased in patients with coronary artery disease, albeit within a small cohort study [130]. This was complemented by a further study that demonstrated higher TN-C levels in patients with acute coronary syndrome, with a ruptured plaque on emergency percutaneous coronary intervention versus stable angina pectoris [131]. Levels are elevated in patients following MI, peaking at day five [116]. Plasma TN-C levels are higher in patients with heart failure with a preserved ejection fraction (HFpEF), correlating with the disease severity (NYHA classification and brain natriuretic peptide level) and markers of interstitial fibrosis. In a multivariable analysis, TN-C was independently associated with adverse outcomes (all-cause death, heart failure hospitalization) [132]. In HF patients, reverse LV remodeling following cardiac resynchronization therapy was associated with a significant reduction in TN-C levels [133]. The levels also correlated with LA size in patients with AF [134]. TN-C concentrations, on top of traditional risk factors, modestly improved prediction of the risk of all-cause death in patients with type-2 diabetes mellitus [135]. Analysis of tissue-based expression of TN-C in endomyocardial biopsy samples (*n* = 123) demonstrated higher expression in diabetes, association with a lower ejection fraction and decreased survival [136]. TN-C was also an independent predictor of cardiac mortality in dialysis-dependent patients (*n* = 238) [137].

## 8. CCNs

CCN proteins are also integral to diverse biological processes. The CCN acronym is derived from the names of the first three proteins identified within that group: cysteine-rich 61 (CYR61, CCN1), connective tissue growth factor (CTGF, CCN2) and nephroblastoma overexpressed (NOV, CCN3). Three additional members have also been described based on structural domain similarity, which pertain to a conserved cysteine residue for disulfide bond formation. These are known as WISPs 1–3 (Wnt-inducible secreted proteins) and are denoted CCN 4–6, respectively. CCN1 [138] and 2 [139] share a redundancy in vascular development, with expression in endothelial and vascular smooth muscle cells during embryogenesis, and homozygous global knockout mice show embryonic or early postnatal mortality [138,140].

The majority of the literature has focused on the role of CCN1 in vascular disease and limiting restenosis after vascular intervention [141], the angiogenic capacity of CCN2 [142] and the anti-proliferative capacity of CCN3 [143]. However, further limited evidence suggests that CCN1 modulates the immune response to autoimmune myocarditis [144], with upregulated expression in response to myocardial infarction or pressure overload, which is recapitulated in tissue from patients with end-stage ischemia cardiomyopathy [145]. Similarly, CCN2 is found in cardiomyocytes following acute MI [146] and in fibroblasts and endothelial cells in post-MI heart failure [147] and aortic aneurysms [148].

CCN1 is upregulated in response to hypoxic stress in both endothelial and vascular smooth muscle cells (SMC) [149]. Recombinant CCN1 suppressed hypoxia-induced contraction and proliferation of pulmonary artery (PA) SMC, potentially contributing to the pathogenesis of pulmonary hypertension (PH). Administration of recombinant protein decreased the RV pressure, suggesting a protective role [150]. Increased CCN1 has been demonstrated in patients with PH [151]. The RV outflow tract is subject to strong pro-inflammatory and profibrotic remodeling transcriptional responses in chronic pulmonary emboli. Microarray analysis of RV transcriptomic changes, and subsequent gene ontology and KEGG analyses, indicated a significant decrease in genes involved in cellular respiration and energy metabolism, as well as increases in inflammatory cell adhesion molecules and extracellular matrix proteins [152]. Signal pathways for wound healing such as fibroblast growth factor, collagen synthesis and CCN proteins were strongly upregulated.

In analyses of LV disease, CCN1 levels were predictive of the three-month mortality (*n* = 248 patients) in patients with acute heart failure [153]. This supported an earlier study demonstrating elevated CCN1 levels in patients with acute heart failure, correlating with pro-BNP levels and a lower six-month survival [154]. CCN1 has been shown to regulate fibroblast senescence, and was overexpressed following apical cardiac resection in the early postnatal phase. Adenoviral knockdown of CCN1 demonstrated fewer senescent cells in conjunction with suppressed cardiomyocyte proliferation. Injection of recombinant CCN1 following MI resulted in greater numbers of senescent cells and fewer proliferating fibroblasts, with a reduced infarct size and improved LV ejection fraction [155].

CCN2 protects the ischemic and pressure-overloaded myocardium from pathological remodeling. In a model of ischemia-reperfusion injury, transgenic mice with cardiac-restricted overexpression of CCN2 that were subjected to transient occlusion of the left anterior descending (LAD) coronary artery displayed a significantly reduced infarct size [156]. When subjected to permanent ligation of the LAD, mice had attenuated LV remodeling and improved LV function despite having similar infarct sizes compared to controls [157]. In response to chronic pressure overload induced by abdominal aortic banding, the same transgenic mice showed a blunted hypertrophic response and sustained systolic function, without a significant impact on myocardial collagen deposition [158]. Isolated, transgenic mouse hearts perfused with recombinant human CCN2 were similarly protected from ischemia-reperfusion injury [156] through activation of reperfusion injury salvage pathways. Mice subjected to thoracic aortic constriction and treated with CCN2 monoclonal antibody demonstrated significantly preserved LV systolic function but without attenuation of interstitial fibrosis [159]. However, a CCN2 monoclonal antibody had no effect on LV remodeling in mice treated with ANGII.

## 9. Neglected ECM Components

Vitronectin is an acute-phase glycoprotein that interfaces with the plasminogen activation/plasmin (PA) system by binding plasminogen activator inhibitor 1 (PAI-1) and blocking vitronectin-integrin interaction. This mechanism limits the development of cardiac fibrosis in response to ANGII infusion, possibly by changing the apoptotic activity and adhesive capacity of fibroblasts [160]. Vitronectin-null mice subjected to MI demonstrated reduced LV dilatation, preservation of LV function and reduced scar formation compared to control mice [161]. Plasma vitronectin receptor levels correlate with the severity of coronary artery disease [162] and have also been shown to be an independent predictor of adverse cardiovascular outcomes following acute coronary syndrome [163] and coronary artery stenting [164]. Vitronectin is also a receptor for osteopontin [165], although whether this binding specifically modulates the cardiac phenotype remains under-investigated.

Perlecan (heparan-sulfate proteoglycan 2) is a long, modular, multi-functional protein crucial to mesenchymal tissue development. Loss-of-function is associated with embryonic lethality due to mechanical instability of the cardiac chamber walls (causing hemopericardium) and loss of basement membrane integrity, causing loss of cellular attachments [166]. Perlecan is expressed by fibroblasts, myofibroblasts and surviving myocytes following MI [167]. Heterozygous perlecan-deficient mice demonstrate reduced cardiac function following MI [166]. ECM influences the outcomes of atherosclerotic disease, although its role varies in mammals. Perlecan is downregulated in human atherosclerosis [168] but is the predominant proteoglycan in atherosclerotic lesions in mice [169]. Apolipoprotein E (ApoE)-null mice crossbred with perlecan-deficient mice demonstrate decreased aortic atherosclerotic lesions, potentially through changed lipoprotein retention and vascular permeability [170].

Syndecans are type-1 transmembrane heparan sulfate proteoglycans, with four family members described. Proteolytic cleavage of the extracellular domain (known as shedding) is precipitated by matrix metalloproteinases and members of the A Disintegrin and Metalloproteinase with Thrombospondin type-1 motifs (ADAMTS) family. This may limit signal transduction or convert the shed domain into a soluble agonist/antagonist. Syndecan 2 (cardiomyocytes) and 3 (fibroblasts) are predominantly expressed in the adult heart [171]. The mRNA levels of all described syndecans are upregulated following MI [172,173], although syndecan-1 [174,175] and -4 [176,177] crucially protect against post-injury cardiac dysfunction through infarct healing. Mice lacking syndecan-1 demonstrate increased leukocyte infiltration within the injured myocardium, associated with upregulated monocyte chemoattractant protein-1 expression and MMP 2/9 activity, and accompanied by collagen fragmentation. These effects were mitigated by adenoviral delivery of syndecan-1 [174]. Paradoxically, loss of syndecan-1 expression protects against ANGII-induced cardiac fibrosis, limiting CCN2 and type-1/3 collagen production [178]. In vitro, this correlated with decreased Smad2 phosphorylation in fibroblasts. A study of plasma syndecan-1 levels (*n* = 567) demonstrated a positive correlation with all-cause mortality and rehospitalization for heart failure in patients with a preserved ejection fraction [179]. Elevated syndecan-1 levels are also independently associated with increased mortality after ST-elevation MI [180].

Absent syndecan-4 increases the susceptibility to cardiac dysfunction and rupture post-MI [181]. This correlates with impaired wound healing, reduced ingress of inflammatory cells and fibroblasts, diminished elaboration of chemokine and cytokines, limited neoangiogenic capacity and decreased ECM deposition. Syndecan-4 is crucial to basic fibroblast growth factor-dependent signaling in endothelial cells that facilitate proliferation and tube formation. Syndecan-4-null cardiac fibroblasts demonstrate reduced migration, as well as lower rates of SMA-positive myofibroblast differentiation and adhesion complexes. Syndecan-4-null mice also show cardiomyocyte apoptosis following MI, leading to increased infarct size, elevated cleaved caspase3 expression and reduced phospho-ERK expression [177]. These findings were associated with cardiomyocyte hypertrophy at the infarct border, due to translocation of the nuclear factor of activated T cells (NFAT). In surviving mice, lack of syndecan-4 changed the LV geometry in healing hearts, improving the ejection fraction at seven days post-injury and limiting increases in end-diastolic and end-systolic volumes compared to control (wild-type) mice. Altered NFAT signaling in cardiomyocytes adjusts the response to pressure overload, and null-mice display LV dilatation rather than hypertrophy [182]. Intact syndecan-4 also protects osteopontin (OPN) from thrombin-mediated cleavage (which also contains a heparan sulfate-binding domain), thereby reducing collagen production in cardiac fibroblasts [65]. Expression of both syndecan-4 transcript and the shed extracellular domain is increased in the myocardium of patients with heart failure [183].

## 10. Conclusions and Future Directions

There is a substantial body of work implicating a broad range of ECM proteins in rodent models of cardiac remodeling following injury (Table 1), with increasing insight into their contribution to the onset, progression or recovery from cardiovascular disease in human trials (Table 2). Their role is structural and yet also involves signaling and interacting with each other and a variety of integrins, growth factors, cell surface receptors and ECM-associated molecules of varying affinities. While these proteins demonstrate significant overlap of function within the myocardium, particularly during repair following acute injury (MI), inflammation (myocarditis) and mimics of chronic disease (pressure overload, aging), there are dichotomous findings between investigator groups. This suggests that post-translational modifications and/or spatiotemporal differences in ECM protein expression may be relevant to biological function, and these aspects have been less rigorously tested in animal models. There are also discrepant findings regarding the role of ECM proteins in pathologies affecting the left and right ventricle. This difference has been illustrated in the context of thrombospondin-1. Overall, there remains a paucity of research into ECM proteins in RV disease, and further work is required to determine whether this is a function of differences in expression between the cardiac chambers and/or disease models. Deciphering the biology of the ECM will enable the development of effective treatment strategies. However, further work is required to determine their clinical therapeutic potential. Many proteins differ in their cellular origins and secretion patterns (development versus homeostasis versus injury), and they also share a functional redundancy, such that a combinatorial approach targeting multiple ECM proteins and/or enzymes responsible for their proteolysis may be required to modify disease-specific or reparative processes.

## Figures and Tables

**Figure 1 ijms-23-01338-f001:**
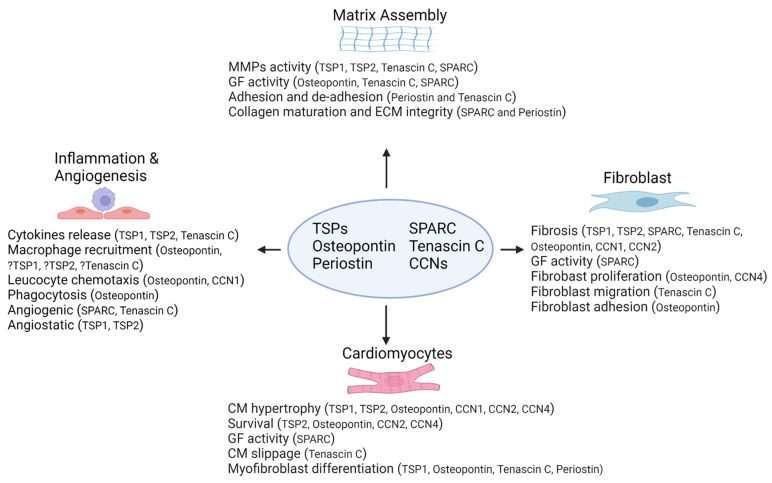
Regulation of cellular function and ECM organization in cardiac pathology. MMPs: matrix metalloproteinases; GF: growth factor; ECM: extracellular matrix; SPARC: secreted protein that is acidic and rich in cysteine; TSPs: thrombospondins; CM: cardiomyocytes; CCNs: cysteine-rich protein 61(Cyr61, CCN1), connective tissue growth factor (CTGF, CCN2) and nephroblastoma overexpressed protein (Nov, CCN3).

**Table 1 ijms-23-01338-t001:** Matrix proteins in murine experimental models of cardiac pathology.

Matrix Protein	Role in Non-Ischemic Cardiac Pathology	Role in Ischemic Cardiac Pathology	Proposed Molecular and/or Cellular Mechanisms	Ref.
TSP1	TSP1-null mice + TAC banding lead to LVH and dilatation. TSP1-null mice + hypoxia (PAH model) have less RVH and arteriolar thickening.	Upregulates expression in infarcted heart (especially border zones), limiting extension of fibrosis into non-infarcted zones.	- Inhibits MMP-2, -3 and -9 activity - Regulates expression of pro-inflammatory cytokines/chemokines - Controls macrophage and myofibroblast infiltration into infarcted myocardium - Regulates TGF-β activity - Maintains sarcolemmal integrity	[9,10,13,14,15]
TSP2	Pressure overload model TSP2-null mice + ANGII infusion lead to mortality from hemorrhagic cardiac rupture and pericardial tamponade. Aging TSP2-null mice + age develop impaired systolic function, cardiac dilatation and fibrosis compared to control mice. Viral myocarditis TSP2-null mice + human coxsackievirus B3-induced viral myocarditis experience increased cardiac inflammation, injury and mortality.	TSP2 may protect structural integrity of myocardium post-MI.	- Regulates MMP-2 and -9 expression - Regulates activity of anti-inflammatory T regulatory cells - Activates the Akt survival pathway in cardiomyocytes	[10,18,19,20]
TSP3	TAC banding = Upregulated myocardial TSP3 expression. TSP3 cardiomyocyte-specific overexpression + TAC lead to exaggerated hypertrophy and fibrotic response and greater cardiac dysfunction.	No known role.	- Limits integrin expression and promotes sarcolemmal instability	[11]
TSP4	ANGII, AVP, TAC = Upregulated myocardial TSP4 expression. TSP4-null mice + ANGII develop LV hypertrophy, fibrosis and aortic dissection.	MI = Upregulates myocardial TSP4 expression.	- Suppresses deposition of ECM proteins - Modulates expression of pro-inflammatory and fibrotic genes	[12,23,24]
MMPs	ANGII = Increased MMP-2 and -9 expression. Spontaneously hypertensive rats = Increased MMP2. MMP inhibition protects the volume-overloaded heart from dilatation and LV hypertrophy. MMP-2-null mice and mice with transgenic overexpression of MMP-2 are protected from TAC-induced LV hypertrophy.	MI = Upregulates myocardial MMP-1, -2, -3, -7, -8, -9, -12, -14 and -28 expression. MMP-2-null mice and WT mice treated with an MMP-2 (and MMP-9) inhibitor are protected from cardiac rupture following MI.	- Regulates leukocyte infiltration - Upregulates JAK-STAT and NF-κB signaling	[34,35,36,42,46,47,50,51,52]
Osteopontin (OPN)	ANGII, TAC = Increased myocardial OPN expression. OPN-null mice + ANGII display reduced hypertrophy, fibrosis and systolic function.	OPN is produced by macrophages within infarcted myocardium. OPN-null mice + MI develop greater cardiac dilatation. OPN-null mice + repeated IRI develop small non-transmural infarctions and ventricular dysfunction.	- Promotes signaling via p38 MAPK and c-Jun - Regulates adhesion of cardiomyocytes to ECM proteins - Induces collagen I expression by cardiac fibroblasts - Modulates expression of ECM proteins, including TN-C, and MMPs	[65,66,67,70]
Periostin	TAC = Increased myocardial expression of periostin. Periostin-null mice + TAC develop less hypertrophy and fibrosis.	Periostin is re-expressed by cardiac fibroblasts following MI and protects infarcted myocardium from ventricular wall rupture. Periostin-null mice + MI have reduced fibrosis and scar formation compared to controls (with survival after acute injury).	Alters fibrotic gene programming in cardiac fibroblasts	[88,89]
SPARC	TAC = Increased myocardial expression of SPARC. SPARC-null mice + TAC have altered collagen processing and reduced diastolic dysfunction. SPARC-null mice + viral myocarditis have increased mortality.	SPARC-null mice + MI have preserved LV function compared to controls BUT demonstrate increased risk of cardiac rupture, HF and mortality. Adenoviral overexpression of SPARC+ MI prevents cardiac dilatation and dysfunction.	- Regulates macrophages, regulatory T cells and leukocytes - Regulates proliferation and migration of cardiac stromal cells - Mediates post-synthetic collagen processing and interaction with cardiac fibroblasts - Alters expression of ECM and adhesion molecule genes in fibroblasts - Activates TGF-β signaling pathway	[97,98,99,100,101,104,111,112,113,184]
Tenascin C (TN-C)	TN-C-null mice + TAC or ANGII show attenuated hypertrophy and fibrosis with preserved cardiac function. TN-C-null mice + ANGII treatment or abdominal aorta constriction have reduced cardiac function.	MI = Increased TN-C expression by cardiac fibroblasts at border zone between necrotic and intact myocardium. TN-C-null mice + MI have higher LVEF compared to WT mice. Transgenic overexpression of myocardial TN-C + MI has higher mortality rates.	- Modulates M1/M2-macrophage polarization - Modulates adhesion between cardiomyocytes and ECM following MI - Regulates pro-inflammatory signaling pathways - Dendritic cell activation and Th17 cell differentiation in autoimmune myocarditis	[120,121,122,124,125,126,127,128,129]
CCN1	ANGII, adrenergic stimulation = Increased CCN1 myocardial expression. CCN1 may be protective against PAH and autoimmune myocarditis.	MI = CCN1 expression is upregulated in ischemic and remote LV myocardium.	- Regulates fibroblast senescence - Modulates immune cell migration	[144,145,150,155]
CCN2	Transgenic mice with cardiac-restricted CCN2 overexpression + abdominal aortic banding have a blunted hypertrophic response and sustained LV systolic function compared to controls.	MI = CCN2 expression is upregulated in non-ischemic myocardium in rats. Transgenic mice with cardiac-restricted overexpression of CCN2 + MI have reduced infarct size, attenuated LV remodeling and improved LV function compared to controls.	- Stimulates fibroblast proliferation - Activates reperfusion injury salvage kinase (RISK) pathways - Reduces hypertrophic signaling pathways	[147,156,157,158]
Vitronectin	Vitronectin binding with PA1-1 and blockade of vitronectin-integrin interaction + ANG II protect against cardiac fibrosis.	Vitronectin-null mice demonstrate smaller infarcts, less ventricular dilation and preserved EF following MI.	Changes apoptotic activity and adhesive capacity of fibroblasts	[160,161]
Perlecan	Perlecan-null mice x APOE-null mice have decreased aortic atherosclerotic lesions.	MI = Perlecan is expressed by fibroblasts, myofibroblasts and surviving myocytes. Perlecan-deficient heterozygous mice develop reduced heart function following MI.		[166,167,170]
Syndecans	Syndecan-1-null mice + ANG II have reduced cardiac fibrosis and dysfunction compared to controls. Syndecan-4 expression is upregulated in the pressure-overloaded myocardium. Syndecan-4-null mice develop increased cardiac dilatation and dysfunction compared to WT mice.	MI = Upregulated syndecan (1–4) expression. Syndecan-1-null mice + MI display exaggerated cardiac dilatation and failure. Overexpression of syndecan-1 via adenoviral gene expression protects against HF post-MI. Overexpression of myocardial syndecan-4 + MI demonstrates less fibrosis and mortality, and improves cardiac function. Syndecan-4-null mice + MI experience greater myocardial injury, enhanced hypertrophic response and increased risk of myocardial rupture. Surviving mice show improved EF at seven days post-MI.	Syndecan-1: - Inhibits leukocyte adhesion and migration - Downregulates monocyte chemoattractant protein-1 expression - Regulates MMP-2 and -9 activity - Regulates Smad2 phosphorylation Syndecan-4: - Regulates cardiac fibroblast signaling, adhesion, migration and differentiation - Regulates cardiomyocyte apoptosis - Alters caspase3 and phospho-ERK expression, along with NFAT signaling - Regulates collagen production in cardiac fibroblasts	[173,174,175,176,177,178,181,182,183]

Abbreviations: ANGII: angiotensin II; ERK: extracellular signal-regulated kinase; LV: left ventricle; LVH: left ventricular hypertrophy; MI: myocardial infarction; MMPs: matrix metalloproteinases; NFAT: nuclear factor of activated T cells; PAI-1: plasminogen activator inhibitor 1; PAH: pulmonary arterial hypertension RV: right ventricle; SPARC: secreted protein that is acidic and rich in cysteine; TAC: transverse aortic constriction; TN-C: tenascin-C; TSP: thrombospondin; WT: wild-type.

**Table 2 ijms-23-01338-t002:** Matrix proteins in human cardiac pathology.

Matrix Protein	Role in Diagnosis of Human Cardiac Pathology	References
TSP1	TSP1 is upregulated in lung parenchyma and vasculature in PH. Plasma TSP1 levels are increased in patients with PH (*n* = 93) compared to controls without PH (*n* = 19). Higher plasma TSP1 levels correlate with mean pulmonary artery pressure and reduced survival. An SNP in the coding region of TSP1 (N700S) confers an >8-fold increased risk of MI in Caucasians.	[16,17]
TSP2	In patients with aortic stenosis, TSP2 expression is increased in patients with LVH and reduced EF (*n* = 5) compared to patients with LVH and preserved EF (*n* = 20). In patients with HF (*n*- = 188), high TSP2 levels correlate with HF-related death and all-cause mortality. In patients with HFpEF (*n* = 150), high plasma TSP2 levels are independently associated with cardiovascular events and risk of death. A TSP2 variant (SNP in 3′ untranslated region) is protective against MI.	[10,21,22,28]
TSP3	Not yet studied.	
TSP4	A TSP4 variant (A378P), present at a high frequency in Caucasian populations, is associated with a lower rate of MI.	[26,27]
MMPs	MMP-1, MMP-2 and MMP-9 levels vary in a time-dependent fashion post-MI and correlate with cardiac function. Plasma MMP-9 levels predict CV mortality in patients with CAD (*n* = 1127).	[38,40,41]
Osteopontin (OPN)	High OPN levels in patients with aortic stenosis (*n* = 149) may be predictive of LVH reversibility following surgical aortic valve replacement. Plasma OPN levels correlate with CAD burden and cardiometabolic risk in diabetes mellitus. Angiotensin II receptor blockade (*n* = 94) and co-therapy with an HMG CoA reductase blocker (*n* = 190) reduce OPN levels.	[75,76,77,79,80]
Periostin	Periostin is upregulated in valvular heart disease, HF and following MI.	[93,94,95]
SPARC	In patients with moderate-severe HF (*n* = 154), SPARC levels predict HF-related death, all-cause mortality and risk of recurrent HF-related hospitalization. Small studies have shown plasma SPARC levels to be significantly elevated in patients with coronary artery disease compared to age- and BMI-matched or sex-matched controls. However, they have not been shown to be a sensitive marker of acute MI.	[105,106,107,108,109]
Tenascin C (TN-C)	Serum TN-C levels are increased in patients with coronary artery disease (*n* = 60) compared to controls (*n* = 20). In patients with ACS, TN-C levels are significantly higher in patients with ruptured plaque (*n* = 23) compared to those with non-ruptured plaques (*n* = 29). Serum TN-C may be predictive of LV remodeling and MACE in patients with acute MI. TN-C levels are higher in patients with HFpEF (*n* = 130) compared to age- and sex-matched controls (*n* = 42) and are independently associated with the composite of all-cause death and HF hospitalization. TN-C levels correlate with LA size in patients with AF. Higher myocardial TN-C expression measured in endomyocardial biopsies (*n* = 123) of patients with DCM is associated with reduced EF and decreased survival. TN-C levels are an independent predictor of mortality in individuals receiving hemodialysis (*n* = 238) compared to healthy controls (*n* = 25).	[116,123,130,131,132,134,136,137]
CCN1	There is robust expression of CCN1 in cardiomyocytes of patients with end-stage ischemic cardiomyopathy (*n* = 5) compared to controls with non-failing LVs (*n* = 4). In acute HF, CCN1 levels are predictive of three-month (*n* = 248) and six-month mortality (*n* = 183). CCN1 is increased in patients with SLE + PAH (*n* = 54) compared to healthy controls (*n* = 54), and patients with SLE and no PAH (*n* = 52).	[145,151,153,154]
CCN2	CCN2 levels in patients with acute ST-elevation MI admitted for percutaneous coronary intervention (*n* = 42) are associated with reduced infarct size and improved LVEF at one year.	[157]
Vitronectin	Vitronectin levels are increased in patients with ACS (*n* = 62) compared to healthy controls (*n* = 18) and correlate with severity of CAD. Vitronectin independently predicts MACE following ACS (*n* = 62) and coronary artery stenting (*n* = 238).	[162,163,164]
Perlecan	Not yet studied.	
Syndecans	In patients with HF (*n* = 567), syndecan-1 levels correlate with markers of fibrosis and remodeling. Syndecan-1 levels are associated with an increased risk of all-cause mortality and rehospitalization for HF in patients with HFpEF, but not HFrEF. Syndecan-1 levels independently predict six-month mortality in patients with STEMI (*n* = 206). Syndecan-4 mRNA and protein levels are increased in myocardial biopsies taken from patients with aortic stenosis and hypertrophic myocardium (*n* = 12) compared to controls (*n* = 12). Syndecan-4 expression in the myocardium is increased in failing hearts (*n* = 20) compared to non-failing hearts (*n* = 10).	[179,180,182,183]

Abbreviations: ACS: acute coronary syndrome; AF: atrial fibrillation; BMI: body mass index; CAD: coronary artery disease; CV: cardiovascular; DCM: dilated cardiomyopathy; EF: ejection fraction; HR: hazard ratio; HF: heart failure; HFpEF: heart failure with preserved ejection fraction; HFrEF: heart failure with reduced ejection fraction; LA: left atrial; LV: left ventricular; LVH: left ventricular hypertrophy; MACE: major adverse cardiac events; MI: myocardial infarction; MMP: matrix metalloproteinase; OPN: osteopontin; PAH: pulmonary arterial hypertension; SLE: systemic lupus erythematosus; SNP: single nucleotide polymorphism; TN-C: tenascin C; TSP: thrombospondin.

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
