# Peer review of "The Role of Matrix Proteins in Cardiac Pathology"

_ijms, 2022, doi:10.3390/ijms23031338_

Round 1

Reviewer 1 Report

The authors of the manuscript have analyzed and summarized a substantial body of works regarding the ECM proteins. The presented review is well organized and comprehensively described, and can be published without corrections.

Author Response

We would like to thank this reviewer for their generous comments. 

We have made significant changes to the manuscript in response to Reviewers 2&3.

Reviewer 2 Report

The review by Trinh et al aims to describe new developments in ECM research related to cardiac pathology. However, the review appears as a rather random collection of ECM proteins and descriptions of their behavior in different cardiac injury models. A common theme is not apparent and the reader will learn next to nothing about how this molecules cooperate in the response to various cardiac injuries or coordinate regeneration. The table is helpful, but again it is unclear why these molecules were selected. Finally, the review is written by a team of authors with the primary expertise in kidney injuries and a very narrow understanding of ECM biology is obvious at many places throughout the review. Language editing is recommended.

Main points (incomplete list):

  • The selected ECM molecules need a better justification and the review should aim at synthesizing knowledge about the function of these molecules in cardiac pathology
  • The title should probably read: The role of extracellular matrix proteins in cardiac pathology. The authors clearly describe more than one protein.
  • The abstract is very difficult to comprehend.
  • Lines 35-37: How does embryonic and perinatal lethality demonstrate fundamental role in cardiac morphogenesis? Fibrillin knockout mice die of ruptured aneurysms. The hearts are fine.
  • Lines 99 – 111: Very difficult to comprehend.  

Author Response

We have included our response to this reviewer in the attached Word document.

We appreciate the opportunity to revise our manuscript.

Reviewer 3 Report

The submitted review article comprehensively describes various roles of ECM proteins in a number of cardiovascular pathologies, despite MI being the primary pathology. Also, the title is misleading based on review centered on multiple proteins. Grammar, tense usage, subject-verb agreement, word choice. incomplete sentences (example pg. 9 line 416), lack of defining abbreviations in text and figure legends, misspellings, etc make this manuscript difficult to evaluate. Extensive revision is required.

Scientifically, there are a number of generalizations and conclusions drawn by the authors that are incorrect or not completely understood. Below are some examples.

pg 3, line 69-70: are you referring to a + or - correlation of MAP and TSP1?

pg 3, line 75: what is a standard ANGII infusion?

pg 3, line 79: AAV transfer of what?

pg 6, line 197: -null mice?

There are no call-outs for figure or tables. Formatting of all figures and tables is required. They are very difficult to read. Legends is also required for figure.

Overall, this was a poorly written and somewhat disappointing review. Improvement would greatly enhance the manuscript for future publication.

Author Response

(The authors gave the same response as above.)

Round 2

Reviewer 2 Report

No further comments.

Author Response

We note that the reviewer has written that a final spell-check is required. We have completed an extensive spell- and grammar-check and have maintained use of British spellings eg remodelling, hospitalised. Please advise if you wish this to be changed to American-style spellings.

Reviewer 3 Report

The authors have made significant changes in English and grammar; however, the remaining errors do not distract from the overall goal of the review. The terms "protective" and "damaging" used in the figures can be construed as misleading. In the research articles cited, most were pathological in nature. Terms, such as compensatory, reparative, preventive, ineffective, etc., were used, and therefore, the figures should be reflective of these findings. Inappropriate generalizations regarding findings should be avoided, and figures should be edited to incorporate literature results.

Author Response

We would like to thank the reviewer for their additional comments. 

We have re-read the manuscript and can find no further grammatical errors. Once again we have made use of British-style spellings.

We also believe the reviewer is referring to the Table 1 rather than Figure 1 in describing the potentially misleading nature of our descriptive terms "protective" and "damaging". We agree that we have extrapolated findings from pathological models, rather than from true experimental work that demonstrates the inherent nature of matrix protein biology during homeostasis (which is difficult to prove). Therefore, to avoid confusion we have deleted these sub-headings from Table 1.